# Management of the Organic Fraction of Municipal Solid Waste in the Context of a Sustainable and Circular Model: Analysis of Trends in Latin America and the Caribbean

**DOI:** 10.3390/ijerph19106041

**Published:** 2022-05-16

**Authors:** Leidy Marcela Ulloa-Murillo, Lina María Villegas, Alejandra Rocío Rodríguez-Ortiz, Mónica Duque-Acevedo, Francisco Joaquín Cortés-García

**Affiliations:** 1Faculty of Agrobiology, Food and Natural Resources, Czech University of Life Sciences Prague CZU, 16000 Prague, Czech Republic; 2Faculty of Health, Universidad del Valle, Cali 760043, Colombia; lina.villegas@correounivalle.edu.co (L.M.V.); alejandra.rodriguez@correounivalle.edu.co (A.R.R.-O.); 3Department of Agronomy, Research Centre CIAIMBITAL, University of Almería, 04120 Almería, Spain; mda242@ual.es; 4Department of Economy and Business, Research Centre CIAIMBITAL, University of Almería, 04120 Almería, Spain; 5Faculty of Business and Management, Universidad Autónoma de Chile, Santiago 7500912, Chile; franciscojoaquincortesgarcia@gmail.com

**Keywords:** circular economy, bioeconomy, biowaste, food waste, waste management

## Abstract

The main objective of this research is to analyze the most relevant aspects of the management of the organic fraction of municipal solid waste (OFMSW) and the Sustainable and Circular Production Models (SCPMs) in Latin America and the Caribbean (LAC). The bibliometric method was used for the analysis of 190 studies obtained from the Scopus and Latin America and The Caribbean on Health Sciences (LILACS) databases. The systematic review provided information on the main research approaches: identification and characterization; quantification; strategic and interdisciplinary management; and processes for treatment or valorization. Finally, an evaluation of public policies and strategies was performed. The results show that Brazil, Mexico, and Colombia have the highest number of publications on OFMSW. The findings also indicate that both research and policy strategies on SCPMs prioritize bioenergy and biofuels as the leading alternatives for the valorization of OFMSW. It also reflects the relevance of the Circular Economy (CE) and Bioeconomy (BE) as the main drivers of waste recovery and/or valorization in LAC. These aspects are of great interest to governments that are still in the process of implementing SCPMs. However, for those more advanced in this area, it provides valuable information on progress, policy effectiveness, and future actions for improvement.

## 1. Introduction

According to the UN, Environment Municipal Solid Waste (MSW) in Latin America and the Caribbean (LAC) reached a volume of almost 540,000 tons per day, and the expectation for 2050 is that the solid waste produced in the region will increase by 25% [1]. However, the problem is not exclusively associated with the constant increase in MSW production, but with inadequate management that produces serious economic, social, and environmental impacts [2]. Food waste constitutes the largest share of MSW in LAC (50%); for low-income countries, this type of waste accounts for almost 75%, from which approximately 27% still goes to illegal open dumps or is subjected to burning or other polluting practices [1]. The degradation of the organic fraction of municipal solid waste (OFMSW) generates significant amounts of methane, one of the most polluting greenhouse gases [3].

In general, LAC has a low recycling rate, so about 90% of waste is disposed of in landfills. Additionally, 7% of the LAC population, mainly in rural areas, does not access waste collection services, which generates waste accumulation and, consequently, the proliferation of contaminating vectors [1], which has been linked to vector-borne diseases, such as malaria, eczema, typhoid, and cholera [4]. The World Health Organization (WHO) has reported the adverse health effects associated with inadequate management of solid waste in the population living in residential areas close to waste disposal sites, which are exposed through diverse pathways to the generated pollutants [5,6]. Vinti et al. [7] carried out a systematic review where evidence is reported, supporting that populations living near landfills have a higher risk of respiratory diseases, mortality [8], and potentially influences their state and wellness [9]. Additionally, the high risk of congenital malformations [10,11], lung cancer mortality related to exposure to airborne contamination (H_2_S) [12], and health risks associated with groundwater contamination due to dumpsites leachate has been studied, with a range of potentially toxic elements (As, Cd, Cu, Pb, Ni, Zn) [13].

The main MSW problem in LAC is associated with the traditional collection and final disposal system in landfills. Among the aspects that derive from this type of management are disinterest in separation at source practices, lack of adequate recycling programs in households, lack of waste collection service coverage, chronic littering in the streets, lack of sanitary landfills, and shortage of funds [2,14]. Global Warming Potential (GWP) is also a matter of concern due to the direct influence that it can have on the implementation of different MSW management strategies [15]. In addition, different topics of analysis missing in the region that have been widely studied in Europe have been identified, such as landfill mining [16,17], and the specifications on trace compounds from OFMSW [18].

Although LAC governments have made a major effort to address these challenges in recent years, much work has to be done to achieve the goals of the 2030 agenda [1]. This is mainly because the Sustainable Development Goals (SDGs) targets in this area are ambitious. SDG 11-“Sustainable cities and communities” aims to achieve the following by 2030 “*to reduce negative environmental impact generated in the cities per capita, paying special attention to air quality and the management of municipal and other types of waste*”. Similarly, SDG 12, “Responsible Consumption and Production”, includes among its targets “*significantly reduce waste generation through prevention, reduction, recycling and reuse*” [19]. Reducing food waste and food losses in production and supply chains is another target of this SDG [20].

Undoubtedly, achieving these objectives requires synergies between governments and the different sectors and/or actors involved in MSW management, strengthening waste separation practices, and providing a legal framework aiming to promote waste management and recycling alternatives [21]. Similarly, it is necessary to implement strategies that lead to an alternative and more sustainable development model. In this sense, the circular economy (CE) and the bioeconomy (BE) have become key enabling tools to move away from traditional practices of linear production and consumption [22,23]. These Sustainable and Circular Production Models (SCPMs) prioritize the efficient management of resources and the reincorporation of waste into production processes to avoid losses of valuable materials and reduce environmental impacts [24].

Under the SCPMs approach, the reduction of waste going to landfills is promoted [22]. Similarly, the conversion of waste into valuable products (organic amendments, animal feed, bioenergy, biofuels) contributes to the development of new markets. At the same time, it generates economic incentives and greenhouse gas emission reductions [21]. The organic fraction of MSW constitutes an important raw material in the context of the SCPMs [25]. This represents an excellent opportunity to attract the interest of numerous stakeholders as a result of its potential positive impacts on waste management and economic growth opportunities [23].

According to the World Economic Forum, the shift to a circular model could generate $1 trillion in material savings in the next five years alone, avoid up to 100 million tons of waste, and create more than 100,000 vacancies [26]. Unfortunately, MSW management is not yet seen by the private sector as an investment opportunity [2]. Similarly, although SCPMs have become more critical in recent years in LAC, adopting concrete policies on this issue has been slow. In addition, few countries currently have specific policy strategies on CE and/or BE [23].

In early 2021, LAC’s environment ministers launched the Circular Economy Coalition, a regional initiative to drive the transition to a sustainable economic system, improving knowledge and understanding of CE. In this way, and with access to funding, it aims to support governments in driving the transition to a sustainable economic system [21]. This is an indicator of the relevance of LAC countries as key scenarios for the implementation of SCPMs. Their high potential in biomass production and the availability of large quantities of residual biomass makes the development of sustainable and circular practices in all sectors feasible [27]. All of the above, in addition to the important challenge of improving MSW management in LAC, prompted the development of this research, which has the following among its main objectives:To analyze the main characteristics of the scientific production on OFMSW.To identify the main approaches of studies on the management of OFMSW and their relationship with SCPMs.To evaluate the regulatory framework, policies, and/or strategies on MSW management and SCPMs.

Previous studies have analyzed LAC MSW management and recovery alternatives [2,28]. Similarly, others have focused on assessing trends in adopting the circular economy and bioeconomy in Latin America [29,30]. However, this study aims to address the identified research gap in previous studies, which includes a lack of evaluation of scientific production on OFMSW, identification of the main approaches and research trends, and the inclusion of a scheme of the regulatory framework in LAC to support SCPMs, along with the current barriers. This novel research aims to integrate all these aspects; it offers a joint view of the management of OFMSW and SCPMs based on the analysis of key research and instruments such as public policies, regulatory frameworks, and management strategies. Under this holistic approach, the results of this research are of interest to governments, entrepreneurs, and academics. This is mainly because it provides valuable information on current trends, the incidence and/or contributions of new policies, as well as the improvement actions and/or aspects that should be considered for these countries to advance in the purpose of managing MSW under the SCPMs approach.

## 2. Materials and Methods

### 2.1. Analysis of the Main Characteristics of Scientific Production

#### 2.1.1. Source of Data and Search Strategy

A total of 190 studies were obtained, 161 from Scopus database, which is considered one of the biggest repositories of academic and peer-reviewed publications with more than 84 million records [31], and 29 studies from LILACS, a database that includes scientific research on health sciences specifically for Latin America and the Caribbean [32], which allowed us to get representativeness on the research carried out and published in the region. The research found in LILACS that was already in the Scopus database was eliminated. Table 1 lists the terms and other criteria considered for the search.

Articles were selected as analysis documents to guarantee the validation process provided by peer reviewing, which intends to validate scientific research.

#### 2.1.2. Data Analysis

The analysis of the publications was carried out using bibliometric techniques. This method has been widely used in recent years for reviewing the structure and trends within a research field. Bibliometric variables allow to present a visualization of the scientific scheme in the field, producing qualitative structured literature reviews [33]. This method has widespread use in different areas of knowledge due to its ease and advantages in evaluating scientific literature, such as omics and bioinformatics [34], circular economy framed in the SDGs [35], and agricultural waste scientific research evolution [36]. Among the general variables analyzed are (1.) Evolution of publications. (2.) Countries with the highest number of publications. (3.) Most cited articles, and (4.) Main journals. Knowledge mapping was performed using VOSviewer (V1.6.14), focalizing the “link strength” of networks based on author keywords. We manually edited the thesaurus list to identify emerging terms to exclude irrelevant terms (i.e., using and quantity). We used the fractional counting method to produce the co-occurrence network, with a minimum of two occurrences for each term.

### 2.2. Identification of the Main Research Approaches—Systematic Review

The 190 studies included in the general analysis were reviewed in detail, mainly the title and abstract and, when necessary, other sections. From this, common aspects were identified concerning the focus of the research. These main aspects were defined as grouping approaches, which are: (1.) Identification and characterization of OFMSW, (2.) Quantification of food waste, (3.) Strategic and interdisciplinary management of OFMSW, and (4.) Techniques and/or processes for treatment and/or valorization of OFMSW.

#### Analysis of SCPMs Approaches

We reviewed and selected research that included the terms “circular economy”, “bioeconomy”, “circular bioeconomy”, “sustainability”, and/or “ODS”. The fields analyzed were: title, abstract, and keywords. A total of 32 studies were obtained. Subsequently, these studies’ main reflections and/or contributions in relation to SCPMs were identified.

### 2.3. Assessment of the Regulatory Framework, Policies, and/or Strategies on Waste Management and SCPMs

#### 2.3.1. Identification of Data Sources and Available Information

The public and/or private entities and key actors with competence in the subject and/or involved with the topic were evaluated and defined. Subsequently, the websites of these institutions were consulted to obtain documents related to the regulatory framework, policies and/or strategies for waste management and SCPMs. In cases where insufficient information was found, e-mails were sent with direct queries. The main entities from which data was obtained include Ministries of Environment and Sustainable Development, Ministries of Planning and Development, and public sanitation service operators.

#### 2.3.2. Revision, Classification, and Analysis of Documents

The regulations, policies, strategies, reports, and other documents related to waste management and SCPMs in LAC were analyzed in detail. From this, the main approaches of these management instruments were identified, and the correlation with published studies on the subject was analyzed. This analysis was carried out for the countries for which scientific publications were recorded.

## 3. Results and Discussion

### 3.1. Main Characteristics of Scientific Production

#### 3.1.1. Evolution of Scientific Production

Figure 1 shows the annual variation of publications in both databases. The earliest publications are from 1993. The Scopus database records a higher number of studies per year. Only in one year (2007) were more studies recorded in the LILACS database. An annual average of 7.3 publications are recorded in Scopus and 1.5 in LILACS. The years 2017 and 2021 are the years with the highest number of publications, with 6 in LILACS and 34 in Scopus. Since 2015 there has been a constant growth in the number of publications in the Scopus database.

Sixty-three percent of the studies were published in the last five years. We observed the evolution of the research topics, which started with composting as the main approach in 1993 publications, moving towards waste to energy, food losses quantification, and strategic and interdisciplinary management in the publications made in the last two years. This could be an indicator of the relevance of the SDGs as drivers of policies and/or strategies that encourage research and/or adoption of practices on the use and/or valorization of OFMSW. This is in line with similar research findings that indicate that in the last ten years, there has been an increase in the number of research studies on the use of agricultural waste biomass [36]. This is also due to new policy guidelines on SCPMs that have given guidance on the potential of waste biomass as a feedstock for new bioproducts [37].

#### 3.1.2. Publications by Country

Three countries lead the ranking of OFMSW publications (Figure 2). Brazil has the highest number of articles in both databases, 43% in Scopus and 79% in LILACS. Similar research shows that Brazil is also the South American country with the highest number of publications on agricultural residual biomass [36]. Additionally, it was found that co-authorship among the retrieved articles was shared with Spain (10 articles), Germany (7), the USA (5), Italy (4), Portugal (4), and the UK (4).

Brazil and Argentina are among the leading countries in bioenergy production, which means they are particularly interested in using waste as the main raw material [23]. Brazil is also one of the leading ethanol producers globally, which demands more significant use of different types of organic waste to obtain this type of product [36]. On the other hand, in Mexico, in the last five years, progress has been made in the development of programs and projects on the energy use of urban waste through cooperation agreements with European countries [3,14]. This may be contributing to the increase in research in this country.

#### 3.1.3. Most Cited Articles

Table 2 shows the top ten cited articles from the search, along with the number and the name of the journal, country, number of authors, and affiliated institutions. The five publications from Scopus represent 77% of the total number of citations in the sample analyzed. López-Torres and Espinosa-Lloréns’ article from 2008 is the most cited, with 175 citations [38]. The second most cited is García Peña et al., with 170 citations [39]. Brazil led the research with three articles co-authored with Italy, followed by the UK. The five publications in the LILACS database represent 23% of the sample. Brazil also led the research with six articles, followed by Mexico. Pranzetti Barreira et al. is the most cited article with 73 citations, published in 2006 in the Brazilian journal Engenharia Sanitária e Ambiental. The number of authors varies from 1 to 7, with an average of 3.6 authors and 2.0 affiliated institutions per article.

#### 3.1.4. Main Journals

Table 3 shows bibliometric information of the five journals with more contributions found in Scopus and LILACS. The British journal Waste Management, categorized by Scimago as Q1 with an H index of 145, had the highest number of articles, 15 in total, followed by the Brazilian journal Hygiene Alimentar, unclassified by Scimago, with 11 publications. These journals have published 32% of the articles in the sample analyzed.

### 3.2. Keyword Analysis

The co-occurrence of keywords network was built using the author’s keywords as criteria. Figure 3 was based on 161 articles retrieved from Scopus; the author’s keywords network was constructed with a total of 379 out of 417 terms (manual edition of the thesaurus list). This figure was assembled in four clusters reflecting the main terms. The resulting network was divided into 4 clusters with 125, 110, 94, and 50 nodes each (379 in total). The nodes with the highest occurrence and link strength were *waste management* (cluster 1), *composting* (cluster 2) central node, and the transition point between all the formed clusters along with *anaerobic digestion*, *biogas*, and *energy* (cluster 3), *recycling*, *sustainability*, and *consumption* (cluster 4).

All these recurrent keywords allowed us to observe that indexed publications are focused mainly on composting as a treatment alternative. This evidence can move the logistics aspects such as selective collection and transport. It is also possible to identify emerging terms in this matter, such as *CE*, related directly to the node *anaerobic digestion*. Some other emerging terms identified in the overlay visualization of this network were *food recovery*, *waste to energy*, and *reuse of FW*, with the average year of publication 2020. The countries that appeared in this network are Colombia, Brazil, Mexico, and Costa Rica, located in cluster 1 (Figure 3).

As a complement, the keyword co-occurrence network for LILACS (Figure 4) was divided into four clusters, having 62 nodes in total. LILACS keywords networks were based on 29 articles; the network was built with 62 out of 85 terms. The most frequent terms are *food services*, *Brazil* (cluster 1), *food handling* and *solid waste collection* (cluster 2), *landfill* (cluster 3), *solid waste*, *anaerobic digestion*, and *recycling* (cluster 4). All these frequent keywords allowed us to observe that the study of OFMSW in the region is focused on strategies for adequate management, the characteristics, and the source of the waste in the included publications.

### 3.3. Main Research Approaches—Systematic Review

#### 3.3.1. Analysis of Publications According to the Classification Category

The research analyzed was grouped into four categories (Figure 5), 41.1% of these publications were focused on techniques and/or processes for treatment of OFMSW, Food losses determination was the second main category (21.6%), OFMSW identification and characterization was in third place (20%), with Strategic and interdisciplinary waste management in fourth place (17.4%).

#### 3.3.2. Identification and Characterization of OFMSW

Regarding the composition of OFMSW, Brazilian authors such as Menezes et al. [48] and Silva et al. [44] characterized household solid waste from different cities in Brazil. In Brazil, Menezes et al. [48] reported the gravimetric characterization of household solid waste from Juiz de Fora. The results indicated that OFMSW corresponds to 43.8%, and the recyclable fraction is 31.7%. The higher socioeconomic status has been associated with a higher generation of recyclable waste [49].

Likewise, Silva et al. [44] explored the relationship between socioeconomic and demographic factors in the production of household solid waste. The authors identified an “urban-contemporary demographic profile”, which reported higher solid waste generation per capita. Conversely, low-income regions showed a higher organic fraction. On the other hand, the Brazilian study by Juffo et al. [50] focused on the characterization of OFMSW produced by restaurants. Regarding the segregation level, only 46% of the collected samples reflected a suitable segregation method; it was found that this aspect could be related to the waste handlers.

Different studies have focused on quantification, characterization, and evaluation of the management and disposal of fruit and vegetable waste (FVW) in supermarkets [51] and educational institutions, where daily waste production is roughly 3.3 metric tons, 52% of this material may be composted, 27% recycled. Just 21% should be disposed in a landfill [52]. Assessing the potential of fruit and vegetable waste (FVW) was addressed by Angulo et al. [53]; a composition of 43% fruit 30% vegetables was found, which suggests that FVW might represent a potential resource for bovine diets. Other findings indicate that this sort of waste provides a good source of both insoluble and soluble fiber, as well as nutrients (Fe, P) [45], protein, and antioxidants [54]. According to the regulations issued by municipal governments, these kinds of MSW generators that produce considerable volumes of solid waste are required to follow waste management programs.

#### 3.3.3. Food Losses Quantification

Food waste embodies not only environmental but also socioeconomic costs related to moral concerns. According to FW Index Report (2021), globally, in 2019, around 931 million tons of food waste was produced by households (61%), food services (26%), and retail (13%). In the study made by Dal’ Magro and Talamini [55], the authors estimated the magnitude of food loss and waste generated in different commodity groups along the Brazilian Food Supply Chain; according to the authors, most food loss and waste were identified during pre-consumption, followed by the consumption stage.

In Brazil, FVW is about 1.4% of the gross domestic product [56]. One of the difficulties associated with FVW production is the absence of a policy framework to decrease losses and promote food donation through food banks or other alternatives [57]. Food loss investigation has been a topic with increasing interest after 2000; however, more data and scientific research are needed, including studies in nutrition units, different studies have quantified food losses, reporting different average values ranging from 5.1% [44], 19.2% [58], 13% [59], to 19% [60]. Losses of fruit and vegetables which were produced but did not get to human consumption as a final destination were determined by Fehr and Romaáo [61]. The total FVW calculated was around 16.6% in the marketing stage.

Waste handling has been identified as a critical issue. It has been analyzed from different approaches, including the nutritional approach [62,63]; its main findings suggest that recycling and avoiding waste during food preparation could be the principal actions to minimize solid waste production. Additionally, some authors aimed to arouse readers’ interest in the opportunity to promote more efficient, sustainable, and inclusive agri-food systems, resulting in a production of excellent quality and promoting responsible food consumption [64].

In this sense, some initiatives have been implemented; for instance, in 2015, the Agri-Food Directorate of Agroindustry Ministry of the Argentine Nation conducted the first exercise to estimate FW in Argentina to analyze the causes, magnitude, and consequences. The research worked on the primary agri-food sectors representing the country’s economic activity in production, exports, and relative importance for regional economies. The work yielded a total volume of FW 16 million tons in its “primary equivalent”, representing 12.5% of agri-food production, where the “losses” explain 90% of the total.

#### 3.3.4. Strategic and Interdisciplinary Management of OFMSW

The implementation of a strategic and interdisciplinary approach for the management of OFMSW could be helpful to face municipal solid waste challenges in the LAC countries; no single model would meet all nations’ needs; each solution must be based on location, physical characteristics, and governmental and cultural factors. It is needed to emphasize the importance of the informal sector during the collection and sorting of MSW, despite the fact it is not incorporated into the official waste management system, which itself contributes to improving MSW operational efficiency [65]. Tools such as education using a door-to-door selective collection program would allow us to enhance the overall waste management system [66]. Additionally, nutritional and environmental education guarantees increasing awareness about the relevance of responsible consumption behavior [67].

#### 3.3.5. Techniques and/or Processes for Treatment and/or Valorization of OFMSW

From the retrieved articles, 44% were focused on techniques and/or processes for treatment and/or valorization of OFMSW, distributed in Composting 41%, waste to energy 34%, anaerobic digestion 18%, and vermiculture 5% (Figure 6).

Among the identified techniques and/or processes for treatment and/or valorization of OFMSW, the retrieved literature was classified into different categories: vermiculture, composting, anaerobic digestion, co-digestion, and waste to energy. It was possible to observe the strong preference of the studies for the usage of composting as an alternative for the treatment of OFMSW, having application also in environmental restoration [68], allowing the reduction of the amount of MSW going to landfills, thus avoiding problems of soil contamination or the emission of harmful gases into the atmosphere. Waste to energy approach was also a primary strategy that enhanced the understanding of the potential of energy production should be determined from OFMSW anaerobic digestion and gas incineration from landfills as an alternative for decentralized energy production within the cities.

(i).Composting

Composting consists of the decomposition of organic materials under aerobic conditions, and it is partially conducted under thermophilic conditions. This type of treatment for OFMSW has been used worldwide [69,70], and some authors have focused on finding advantages of using this type of waste for co-composting of green waste, food waste, and sawdust [71], also co-composting OFMSW and biosolids, the results showed a high-quality compost, with a greater concentration of total nutrient and organic matter [72]. Additionally, benefits associated with a mixture of composts rich in N and P have been found from animal manures with OFMSW on reducing P losses through dilution and liming effects [69]. Small-scale composting has been addressed as a sustainable alternative for households under different climatic conditions [70].

On the other hand, Brazilian studies focused on the use of organic composts derived from the OFMSW in the adsorption of Potentially Toxic Elements (PTEs) [73] and the quality of MSW compost produced in São Paulo [43]. For example, in the study by Lima et al. (2018), the authors found that the evaluated composts followed the removal selectivity order Pb > Cd > Zn, suggesting that composting may be suitable for composting alternative for contaminated soils remediation. Furthermore, composts can be considered soil conditioners because of their low nutrient content [43,74].

(ii).Waste to Energy

In LAC, there is an increasing interest in the academia, national governments, and municipalities on the research, promotion, and use of different alternatives for waste management, such as sorted waste collection, recycling, and waste to energy approach. According to De Souza et al. [75], the biggest cities in Brazil in 2011 generated 18.9 M tons of MSW, of which 51.5% corresponds to OFMSW. These cities reported an energy production capacity of 535.1 MW, of which 43% corresponds to Rio de Janeiro and São Paulo.

Different studies showed anaerobic digestion as a promising alternative for OFMSW valorization on full-scale installations [76]. Additionally, through the determination of its energy potential, performing a comparative analysis for the electrical energy generation using MSW, including methanization of OFMSW in anaerobic digesters, which could guarantee an energy production up to 20.097 GWh per year (equivalent to 0.18 MWh per ton of organic waste) [77]. Biochemical Methane Potential (BMP) test was used to determine a baseline for anaerobic digestion performance, showing anaerobic digestion as an alternative renewable energy source to convert FW into biogas, calculating revenues that can reach up to 106 USD per year [78]. Additionally, BMP has been tested using OFMSW after a water-based extracting method to get soluble substances; the obtained results showed that methane production reached its maximum using the ratio 1:2 (OFMSW: Water) [79].

Green House Gases emissions evaluation and energy consumption models are used to determine the economic benefit that could be obtained from OFMSW management. Pin et al. [80] presented a two-step analysis of the MSW management, allowing the reduction of up to 90,000 tCeq concerning the base scenario and a carbon credit potential income in 1000 USD 242.98. The determination of the calorific value of the OFMSW through gravimetric characterization indicates that this residue is suitable for incineration as an alternative for energy recovery [81].

In Argentina, the study carried out by Morero et al. [82] presented a complete mathematical model for the technology selection of MSW treatment, including co-digestion of SS and OFMSW. The authors aimed to identify the ideal mixing ratios of SS and OFMSW to feed anaerobic digesters. Their results suggested that co-digestion of SS and OFMSW results in a unified waste-to-energy process, magnifying economic benefits and mitigating the environmental impacts of inadequate waste disposal.

Aiming to reduce its greenhouse gas emissions, Mexico has developed a strategy centered on renewable energy production. Furthermore, government institutions have been promoting the transition from diesel to fewer pollutant fuels in the transport sector. Natural gas has been considered a suitable alternative which can prepare the ground for biomethane usage as a transport fuel. Its production from co-digestion of OFMSW and FW has been estimated as a considerable theoretical potential (42.32 PJ per year) [83]. As an alternative, Garcia-Peña et al. [84] evaluated the viability of H_2_ production in batch conditions from FVW sourced from local markets. Hydrogen production from FVW is an innovative and feasible energy technology. Furthermore, constant use of this treatment alternative would allow the transformation of organic residues into bio-energy.

(iii).Anaerobic digestion and co-digestion

The co-digestion approach involves the addition of energy-rich organic waste materials to wastewater digesters with excess capacity. Argentinian authors found that the co-digestion of sewage sludge (SS) and OFMSW has excellent potential for producing biomethane, electricity, and, potentially, fertilizers [85]. On the other hand, the Brazilian study by Proença and Machado [86] focused on the applicability of anaerobic digestion of OFMSW using residential biodigesters. This alternative is suitable for decentralizing the treatment and disposal of OFMSW, including FVW providers such as distribution markets [39].

The Mexican study from García-Peña et al. [39] focused on the feasibility of using FVW from a food distribution market as a substrate for anaerobic digestion. The methane production was possible due to the readily biodegradable and well moist organic matter content of FVW and the use of anaerobic digestion. Moreover, they found that pH control and nitrogen addition improved the anaerobic digestion performance and co-digestion with meat residues allowed to obtain better results.

(iv).Vermiculture

Across the retrieved literature, it is possible to observe that recycling initiatives are still being developed. Still, one technique that has been used is vermiculture, in which certain earthworm species are involved, and they rapidly break down organic material residues further into smaller particles. Torri and Puelles [87] compiled findings from Argentina within the last decade of organic residue vermicomposting and bioaccumulation of potentially toxic elements in artificially contaminated systems. These experiments have demonstrated the viability of including different kinds of waste for vermicomposting; this allowed to transform wastes into a value-added amendment, reducing its economic and environmental impact.

Additionally, few studies used fermentation techniques and/or processes for treatment and/or valorization of OFMSW aiming to produce glucose, ethanol or organic fertilizers [88,89]. Certain factors limit the development of infrastructure for the implementation of different techniques or processes of treatment in developing countries, such as the management of OFMSW and the availability of resources. On a global scale, only 11% of MSW is utilized in waste-to-energy conversion schemes [76]. All the techniques or processes of treatment should incorporate an integrated approach, including geography, socio-economical conditions, technical requirements, and environmental legislation.

### 3.4. Scientific Publications with SCPMs Approach

Table 4 shows studies that include the SCPM concept. The first publication was made by Matharu et al. [42], which offers a bioeconomy approach for food supply chain wastes and valorization as an alternative for global sustainability. Another study evaluated different FW valorization scenarios based on anaerobic digestion and composting compared to landfilling, employing a decision-making setting with a combination of linear programming, analytic hierarchy process, and life cycle thinking [28]. The last article was published in January 2021 [90]. The authors found that, from the environmental scenario, valorization alternatives of FW would lead to reduced global warming potential.

All the articles were published in the last four years, and this is an indicator that policies on SCPMs in LAC have been gradually incorporated in the previous five years. These results are in accordance with previous research showing a similar trend in the years in which research prioritizing the management of agricultural residual biomass under SCPM approaches was published. In contrast, the percentage of publications with this approach is much lower for OFMSWs in LAC [37]. This could suggest that, from the perspective of SCPMs, other types of biomasses such as agricultural residues have a higher priority as feedstock in other countries. In Brazil, for example, waste from the agro-industrial sector has a more significant potential for use in biogas production. Although the advantages of using solid urban waste are also known, its incorporation into the country’s energy matrix is low [93].

### 3.5. Regulatory Framework, Policies, and/or Strategies on Waste Management and SCPMs

#### 3.5.1. Regulatory Framework for Waste Management

Solid Waste Management policies in LAC constitute a baseline. The main objective is to mitigate adverse environmental impacts experienced in the region, including different actions and stakeholders within solid waste management. All countries presented in Table 5 have instruments in this respect. Argentinian government have issued the National Strategy for the Comprehensive Management of Urban Solid Waste 2005–2025 from the Ministry of Health and Ministry of Environment and Sustainable Development [94]. In 2004, National Law No. 25916 on household waste management established the minimum requirements for environmental protection, including aspects such as generation, collection, transport, treatment, and final disposal [95].

In Brazil, research highlights the advances in the approval of comprehensive national solid waste policies, which contribute to reducing waste generation and promoting waste reuse and recycling [36]. The base framework has been stated in Law No. 12.305, which includes the National Solid Waste Policy that brings together a group of guidelines, goals, and actions adopted by the Federal Government regarding integrated management of MSW. This policy includes a systemic approach for solid waste management through different instruments such as selective collection, post-consumer, and reverse logistic systems, also including an interdisciplinary approach with sectoral agreements, scientific research, and environmental education [100]. The inclusion of waste pickers has represented an important challenge to the implementation of this policy [80].

In Colombia, there are multiple policies regarding Waste Management and Recycling. One of them is the National Policy for Integrated Solid Waste Management 2016–2030 which implements the comprehensive management of solid waste as a national directive integrating social, environmental, and health dimensions, contributing to promoting CE, sustainable development, adaptation, and climate change mitigation [111]. The law 1990 issued in 2019 is a policy focused on preventing food loss and waste, incorporating the appropriate context for the initiatives that could be developed to strengthen the food waste reduction programs [109]. Concerning the regulation of the producer’s responsibility on the post-consumer use of containers and packaging (paper, cardboard, plastic, glass, and metal), the Ministry of Environment and Sustainable Development in 2018 issued Resolution No 1407 [109]. Including this type of regulation in the national waste management framework should be regarded as a part of a multidisciplinary approach that seeks the inclusion of the stakeholders, such as manufacturers and consumers of daily use products that could be reincorporated or reused as part of material recovery and sustainable consumption and production models [1].

In LAC, the increasing MSW generation could be regarded as a continuous source of materials that could be reincorporated into the production cycle, as long as different recovery and/or treatment alternatives are prioritized [2]. It has been identified that emerging countries are currently moving towards the switch from final disposal in landfills or open dumpsites to different alternatives for its treatment and/or disposal [122], which can be seen in the evolution of the regulatory framework in the region.

#### 3.5.2. Circular Economy

The purpose of a circular economy is to get the most out of resources while producing the least amount of waste for disposal; this will result in sustainable consumption and production within a system that conserves and optimizes the use of resources [29]. Currently, in LAC, Costa Rica, Colombia, and Mexico have developed a specific legal framework related to CE. Argentina has waste management and recycling policies, but none regarding CE yet; the Provincial Strategic Plan (PEP) from the Ministry of Environment and Sustainable Development, issued in 2005, sets out the guidelines for “Provincial Strategic Plans of Waste Management towards a Circular Economy”.

In several countries, the private sector encourages the adoption of the CE. Particularly, in Argentina, the Association for the Study of Solid Waste (ARS) is leading the private-sector coalition, which has proposed a National Strategy for the Circular Economy, urging the local government to reinforce the policies to provide adequate support for businesses during its transition towards CE. Government ministries in Argentina, Colombia, and Chile have included social innovation programs, such as the National Program for Technology and Social Innovation for Argentina issued in 2013.

Costa Rica has two National Circular Economy Policies at the moment. The National Decarbonization Plan 2018–2050 gives a roadmap for the country to transition to a low-carbon economy, providing mitigation strategies for all sectors of the economy, including provisions around waste management and modernizing the industrial sector through ‘cradle to grave’ product design [107], and the National Policy on Sustainable Production and Consumption 2018–2030 includes objectives around using the CE approach to create sustainable industrial parks and education programs for the manufacturing sector on CE [106]. These programs promote multiple stakeholders to explore FW valorization initiatives, and they can pave the way to support further steps into achieving the SDGs directly.

Regarding the CE policies in Colombia, the National Strategy for the Circular Economy 2018–2022 aims for a new economic development model. This includes maximizing resource value, closing cycles for material, water, and energy, developing new business models, and encouraging industrial symbiosis, among other things. Carbon footprint is reduced through increased efficiency in the production and consumption of materials [112]. Additionally, The National Development plan 2018–2022 has one of its objectives: Implement strategies and economic instruments to make the productive sectors more sustainable, innovative and reduce environmental impacts, with a circular economy approach [123].

Mexico, since 2019, has had the National Zero Waste Vision that intends to transform the traditional waste management model into a CE model. Additionally, Mexico City, a Plan of Action for a Circular Economy, Government of Mexico City 2019 aims for zero waste through a range of strategies such as reducing the amount of packaging, regulation for reducing single-use products, and proper waste management, among others.

Because it allows for intersectoral diversification to develop added value among the materials flow, the circular economy model has gotten a lot of political attention in Latin America in recent years. More than 80 public initiatives focusing on various areas of CE have already been initiated in the region. The development and execution of various national circular economy initiatives have the potential to boost waste recycling rates, which could represent an opportunity for the diversification of economic activities [22,29].

#### 3.5.3. Bioeconomy

Most of the countries in Table 5 have advanced in the development of initiatives related to the bioeconomy. Of the seven countries indicated, Costa Rica and Colombia are the only countries with a specific national policy and/or strategy [108,113]. According to the latest report on global bioeconomy policy by the International Advisory Council on Global Bioeconomy (IACGB) in 2020, Costa Rica is the only LAC country with a bioeconomy policy strategy [23]. This strategy, issued by the Costa Rican government in August 2020, includes among its strategic axes the “*Urban bioeconomy and green cities*”, whose main objective is to promote the application of biological principles in solid waste management. This is why it incorporates, among the lines of action, the sustainable management and valuation of urban solid waste [108].

The plan “*Bioeconomy for a Colombia Living and Diverse Power: Towards a Knowledge-Driven Society*,”, issued by the Colombian government in 2020, proposes a national strategy dedicated to the development of the bioeconomy. This plan prioritizes the use of biomass for the generation of products, processes, and services such as bioenergy. One of the strategic areas and challenges is biomass and green chemistry “Biomass 100”, which aims to provide more value and zero waste. At the same time, it promotes the generation of bioproducts from biofactories and biorefineries [113].

As shown in Table 5, countries such as Argentina, Brazil, and Mexico, although they do not have specific policies, have developed a series of strategic and regulatory instruments related to the bioeconomy. These plans, programs, and/or policies are related to bioenergy, biotechnology, technology, innovation, and continue specific actions related to waste reduction and/or valorization. Moreover, some local governments, such as Buenos Aires (Argentina), have developed specific bioeconomy plans. The IACGB report notes that several of these countries, mainly Argentina, Brazil, and Ecuador, have worked on the development of specific strategies; however, the adoption process has been slow [23].

Brazil was one of the first countries to develop policies related to the bioeconomy [36]. In 2007, it issued the “*Biotechnology Strategy*”, which prioritizes biological processes, the use of clean technologies and innovation in the treatment of industrial, agricultural, and domestic effluents [101]. In Brazil, the bioeconomy has been linked to the bioenergy sector development [23]. For instance, the *Ten-year plan for energy expansion 2020–2029* promotes the use of municipal waste for biogas production. It stresses that this type of waste allows the supply of available resources to be expanded and that benefits can be obtained through its use as biomass in thermoelectric power plants [93].

Mexico’s management and regulatory instruments related to the bioeconomy prioritize actions related to bioenergy. For example, the “*Agreement that approves and publishes the update of the Transition Strategy to Promote the Use of Cleaner Technologies and Fuels*” of 2020 includes among the lines of action the energetic use of urban waste and the recycling of materials, at all levels of government. It also includes the establishment of financing or incentive programs for municipalities and the private sector to use urban waste for energy purposes. One of the lines of action related to social impact seeks to promote the use of rural solid waste to produce biogas through inclusive projects that reduce energy poverty and poverty conditions in general [121].

Some of the documents related to the bioeconomy have emerged due to national projects [97]. Some of them have received international cooperation funds; for example, the “*Program for the Energetic Use of Urban Waste*” in Mexico was implemented in the period 2014–2018 with resources from the German government [14]. This program has enabled the development of several projects aimed at sustainable municipal solid waste recovery by converting it into energy [3].

All the instruments on bioeconomy were analyzed to reflect the relevance of articulated and coordinated work between public and private entities. For example, in Argentina, the 2019 document “*The Bioeconomy as a Strategy for Argentina’s Development*” resulted from a consensus between regions and sectors from different disciplines [98]. Similarly, in addition to being inter-institutional, most of these instruments have been the product of inter-ministerial efforts. Some of them are Ministries of Environment and/or Sustainable Development, Ministries of Science, Innovation and Technology, Ministries of Agriculture, Ministries of Economic, Industry and Commerce, etc. In this same sense, some of the bioeconomy policies highlight their articulation with other public policies, as well as the actors and sectors involved [113]. The articulated work between the government and the private sector in issuing bioeconomy policies in LAC is an aspect highlighted in the IACGB report [23].

It is also important to note that the Economic Commission for Latin America and the Caribbean (ECLAC) has played a fundamental role in developing these bioeconomy programs and policies in LAC. Its role has been relevant both in providing technical assistance on the design of national strategies as well as in the articulation of key actors in the bioeconomy [27]. Similarly, the Latin American Bioeconomy Network created in 2019 has also been key in exchanging experiences, collaborative projects, design of bioeconomy observatories, and in general, the consolidation of the bioeconomy as a regional development strategy in LAC [23].

In general, these instruments prioritize the sustainable use of local resources and the strengthening of capacities to diversify and develop new value chains. The main objective is for the bioeconomy to become the central axis of a new, more sustainable and circular development model. As is evident, many of these instruments prioritize bioenergy and biofuels as the main alternative for the valorization of OFMSW or other types, mainly agricultural residual biomass. This coincides with the findings of previous research [36,37].

## 4. Conclusions

The joint analysis of key research and instruments such as public policies, the regulatory framework, and management strategies provide more comprehensive information to understand this important topic’s evolution, approaches, and trends. This research makes a novel and significant contribution to the scientific literature. On the one hand, it adds new evidence to the relevance that SCPMs policies have acquired in recent years and their influence on the development of scientific research related to the characterization and reduction and/or valorization alternatives for MSW. This is mainly because, although it is evident that the production and management of the organic fraction of MSW is a topic that has been studied in LAC since 1993, it has been in the last five years that this topic has gained greater importance in the field of research. In this sense, it is clear that circular economy and bioeconomy policies and strategies have promoted a more significant interest in reducing and valorizing OFMSW. This is mainly the case in countries such as Argentina, Brazil, Mexico, Colombia, and Costa Rica, which have contributed a more significant number of scientific publications. These countries have also advanced in implementing public policies for integrated MSW management under the approach of sustainable and circular production models. In these countries, anaerobic digestion and composting are the leading treatment technologies used to obtain fertilizers and bioenergy. The latter bio-based product is the valorization alternative prioritized in most LAC MCS policy strategies.

On the other hand, this study reinforces the conclusions of previous government reports and research that highlight the need for further development of SCPMs policies in LAC. It also contributes to the theorization that there is a lower prioritization of OFMSW over other types of waste, such as agricultural waste biomass. This study shows that under SCPMs approaches, OFMSW has not yet been subject to the extensive and varied analysis that other types of waste, such as those derived from the agricultural sector, have been. From a practical point of view, this research’s findings help guide governments on the importance and impact of circular economy and bioeconomy policies and strategies on the management of OFMSW and on the relevance of cross-sectoral and intergovernmental work. Besides, to recognize the need to include in SCPMs policies, lines of action and/or specific programs that further promote the use of OFMSW as a potential raw material for obtaining high added-value products are needed. In the same direction, these SCPMs policies should be focused on research and innovation, which intends to promote the development of technologies and/or processes, such as biorefineries, that would allow the usage of OFMSW to the maximum and expand their valorization alternatives. Finally, the findings of this study also suggest that it is essential to establish funding programs or incentives for industries and/or companies to promote the use of OFMSW in the manufacturing of bio-based products in a local context. In the same vein, international cooperation projects with leading European countries in sustainable and circular production could be essential for less developed countries in Latin America and the Caribbean to advance towards fulfilling sustainable development goals.

## Figures and Tables

**Figure 1 ijerph-19-06041-f001:**
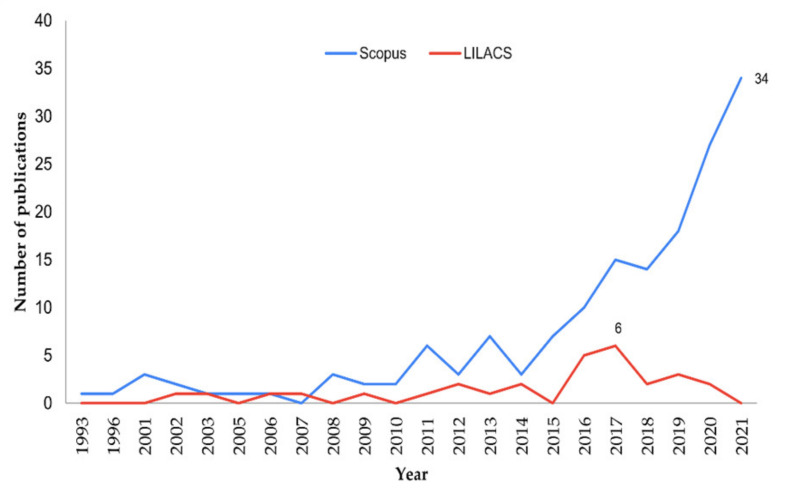
Timeline of publications in Scopus and LILACS.

**Figure 2 ijerph-19-06041-f002:**
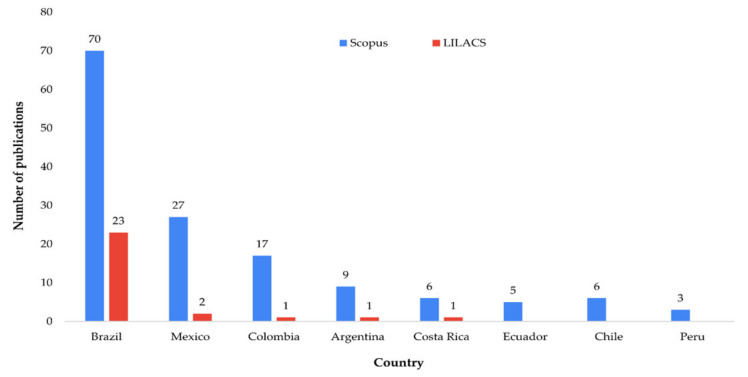
Number of publications indexed in Scopus and LILACS by country.

**Figure 3 ijerph-19-06041-f003:**
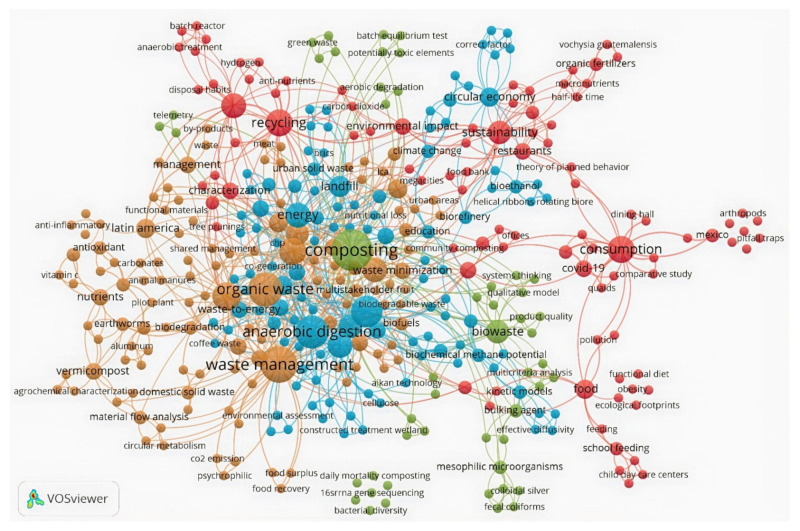
Scopus keywords co-occurrence network.

**Figure 4 ijerph-19-06041-f004:**
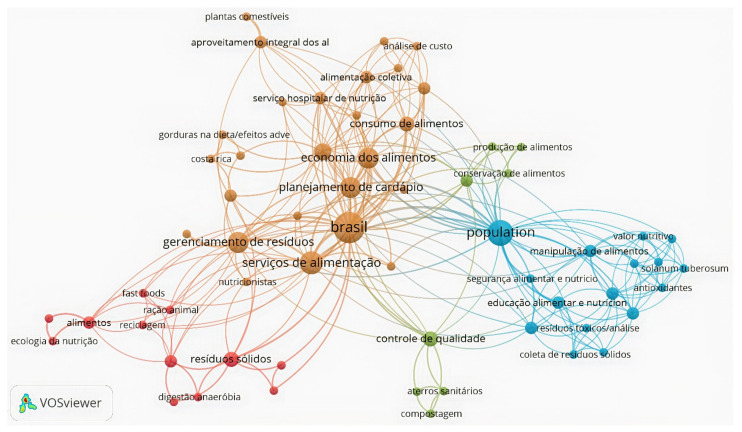
LILACS keywords co-occurrence network.

**Figure 5 ijerph-19-06041-f005:**
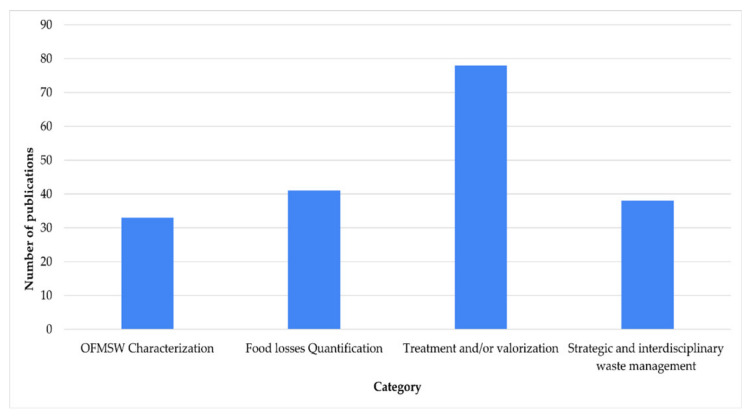
Total of articles published regarding categories of classification.

**Figure 6 ijerph-19-06041-f006:**
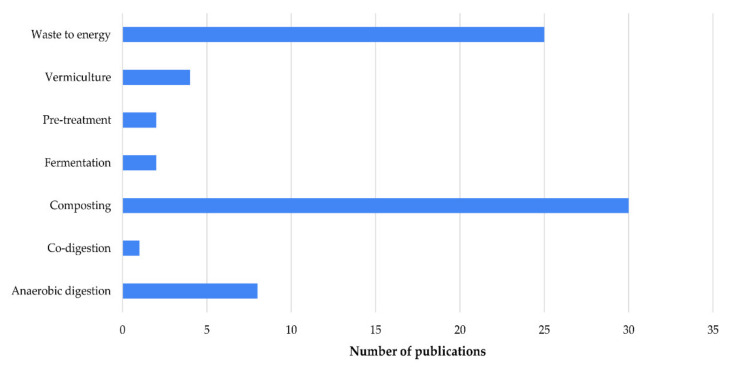
Identified techniques and/or processes for treatment and/or valorization of OFMSW.

**Table 1 ijerph-19-06041-t001:** Main search criteria.

Data Base	Search Equation	Date	Languages	Document Type
Scopus	TITLE-ABS-KEY (“food wast*” OR “household* biowast*” OR “biowast*” OR “biorresidu*” OR (municipal W/5 organic* AND residu*) OR (municipal W/5 organic* AND waste*) OR “organic fraction” OR “OFMSW” OR (kitchen* W/5 waste*) OR (“school canteen*”) OR (“restaurant* waste*”) OR “supermarket* food waste*“ OR (“universit* dining hall*”) OR (fruit* W/5 wast*) OR (vegetable* W/5 wast*)) AND TITLE-ABS-KEY (brazil OR mexico OR colombia OR argentina OR peru OR venezuela OR chile OR ecuador OR guatemala OR cuba OR haiti OR bolivia OR “Dominican Republic” OR honduras OR paraguay OR nicaragua OR “El Salvador” OR “Costa Rica” OR panama OR paraguay OR “Latin America” OR “LAC”) AND (EXCLUDE (LANGUAGE, “Chinese”) OR EXCLUDE (LANGUAGE, “French”) OR EXCLUDE (LANGUAGE, “German”) OR EXCLUDE (LANGUAGE, “Undefined”) OR EXCLUDE (LANGUAGE, “Dutch”)) AND (LIMIT-TO(DOCTYPE, “ar”) OR LIMIT-TO (DOCTYPE, “re”)).	** 10th January 2022 *** All years to 31st December 2021	English Spanish Portuguese	Articles
LILACS	(((food wast*) OR (household* biowast*) OR (biowast*) OR (biorresidu*) OR (municipal organic* residu*) OR (municipal organic* waste*) OR (organic fraction) OR (ofmsw) OR (kitchen* waste*) OR (school canteen*) OR (restaurant* waste*) OR (supermarket* food waste*) OR (universit* dining hall*) OR (fruit* wast*) OR (vegetable* wast*)) AND ((brazil) OR (mexico) OR (colombia) OR (argentina) OR (peru) OR (venezuela) OR (chile) OR (ecuador) OR (guatemala) OR (cuba) OR (haiti) OR (bolivia) OR (Dominican Republic) OR (Honduras) OR (paraguay) OR (nicaragua) OR (El Salvador) OR (Costa Rica) OR (panama) OR (paraguay) OR (Latin America) OR (LAC))).			

** Date of search. *** Date range.

**Table 2 ijerph-19-06041-t002:** Highly cited articles from Scopus and LILACS and metrics list.

Title	Data Base		Year	Journal	Citation *	Authors	Reference
	Scopus	LILACS					
Effect of alkaline pre-treatment on anaerobic digestion of solid wastes	X		2008	Waste Management	175	2	[38]
Anaerobic digestion and co-digestion processes of vegetable and fruit residues: Process and microbial ecology	X		2011	Bioresource Technology	170	5	[39]
Biological treatment of municipal organic waste using black soldier fly larvae	X		2011	Waste and Biomass Valorization	149	5	[40]
The evolution of food donation with respect to waste prevention	X		2013	Waste Management	83	1	[41]
Opportunity for high value-added chemicals from food supply chain wastes	X		2016	Bioresource Technology	67	3	[42]
Composting plants of São Paulo State: compost quality and production processes		X	2006	Engenharia Sanitária e Ambiental	73	3	[43]
Demography of urban consumption: a study on the generation of solid waste in the city of Belo Horizonte		X	2012	Revista Brasileira de Estudos de População	28	3	[44]
Obtention and quantification of fiber dietary some common fruit waste in Colombia		X	2002	Vitae	28	5	[45]
Paradigms of environmental management practices in the meal production sector in Brazil		X	2017	Engenharia Sanitária e Ambiental	23	2	[46]
Comparison of municipal solid waste treatment scenarios through the technique of Life Cycle Assessment: the case of the city of Garibaldi, RS, Brazil		X	2017	Engenharia Sanitária e Ambiental	16	2	[47]

* Citation date 26 January 2022. X indicate that the article was found in the corresponding database.

**Table 3 ijerph-19-06041-t003:** Journals with the highest number of publications.

Journal	NA	SJR	H Index	Country	Subject Categories
Waste Management	15	Q1	161	United Kingdom	Waste Management and Disposal
Hygiene Alimentar **	11	NR	NR	Brazil	Health sciences, nutritional sciences
Revista Internacional de Contaminación Ambiental	8	Q4	19	Mexico	Pollution Waste Management and Disposal
Journal of Cleaner Production	9	Q1	200	Netherlands	Strategy and Management Renewable Energy, Sustainability and the Environment Industrial and Manufacturing Engineering Environmental Science
Engenharia Sanitária e Ambiental **	7	Q3	17	Brazil	Waste Management and Disposal
Sustainability	7	Q1	85	Switzerland	Geography, Planning and Development
Journal of Environmental Management	5	Q1	161	United States	Environmental Engineering Management, Monitoring, Policy, and Law Waste Management and Disposal Medicine (miscellaneous)

NA: number articles; SJR (2020): Q1 journal ranking top 25%, Q2 journal ranking 50–74%, Q3 journal ranking 25–49%, Q4 journal ranking 0–24%. H index: journal’s number of articles (h) that have received at least h citations. ** Articles from LILACS.

**Table 4 ijerph-19-06041-t004:** Scientific publications with a focus on SCPMs.

Title	Year	Journal	Citation *	# Authors	Country FA	Reference
Opportunity for high value-added chemicals from food supply chain wastes	2016	Bioresource Technology	90	3	United Kingdom	[42]
Fruits and vegetable-processing waste: a case study in two markets at Rio de Janeiro, RJ, Brazil	2020	Environmental Science andPollution Research	6	7	Brazil	[91]
Food waste biorefinery advocating circular economy: Bioethanol and distilled beverage from sweet potato	2020	Journal of Cleaner Production	15	3	Brazil	[92]
Decision-making process in the circular economy: A case study on university food waste-to-energy actions in Latin America	2020	Energies	9	5	Italy	[28]
Valorizing municipal organic waste to produce biodiesel, biogas, organic fertilizer, and value-added chemicals: an integrated biorefinery approach	2021	Biomass Conversion andBiorefinery	3	18	Brazil	[90]

* Citation in Scopus 26 January 2022; FA: first author. # Number of authors

**Table 5 ijerph-19-06041-t005:** Main management and regulatory instruments on waste and SCPMs.

Country	Approach			Instrument Name/Year	Objectives/Strategic Lines and/or Specific Measures	R
	MW	CE	BE/CB			
Argentina	X			1. National Strategy for the Integrated Management of Urban Solid Waste 2005–2025. 2. National Law No. 25.916 on household waste management. 3. Provincial Strategic Plan (PEP). 4. Zero Waste Law.	1. The strategy is based on the criteria of integrality, processing and final disposal centers (FDCs). 2. Minimum environmental protection requirements for household waste management. 3. Guidelines for “*Provincial Strategic Plans for Waste Management towards a Circular Economy*”. 4. Prohibits the incineration of municipal waste and provides guidance on recovery and recycling systems.	[94,95,96]
			X	1. The Bioeconomy as a Strategy for Argentina’s Development (2019). 2. Argentina’s Bioeconomy-Vision from Agroindustry (2017). 3. Argentine Biotechnology in the year 2030: Strategic key for a techno-productive development model (2016). 4. Buenos Aires Provincial Bioeconomy Plan (2016).	1. Prioritizes the sustainable use of resources and capacities to diversify and develop new value chains. 2. Articulation between the public and private sectors for an integral development of the bioeconomy. 3. Working agenda to promote sustainable economic and social development. 4. Sustainable agro-industrial development-maximum productive potential based on joint work with all stakeholders.	[97,98,99]
Brazil	X			1. Law N° 12305-National Solid Waste Policy.	Avoidance of landfill disposal. A systemic approach to solid waste management through comprehensive plans.	[100]
			X	1. Ten Year Energy Expansion Plan 2020–2029 (2020). 2. Action Plan on Science, Technology, and Innovation in Bioeconomy (2018). 3. National Strategy on Science, Technology, and Innovation 2016–2022 (2018) 4. Biotechnology Strategy (2007)	1. Integrated vision for the use of diverse energy sources. 2. Promote scientific, technological and innovation development, focusing on sustainable development and social, economic, and environmental benefits production. 3. Collaborative innovation paradigm (universities, companies) oriented towards sustainable development. 4. Development of biotechnology and strengthening of production systems and the national bioindustry.	[93,101,102,103]
Costa Rica	X			1. National Integrated Solid Waste Plan 2010–2021. 2. National Strategy for Waste Separation and Recovery 2016–2021. 3. Law for the Integral Management of Waste No. 8839.	1. Strategies for waste management in public institutions, private sector and social organizations. 2. Inclusive model for integrated solid waste management (public and private sectors). 3. Municipalities’ competencies for the management of generated waste-waste management plans.	[104,105]
		X		1. National Decarbonization Plan 2018–2050. 2. National Policy on Sustainable Production and Consumption 2018–2030.	1. Transition to a low-carbon economy. 2. Mitigation strategies for all sectors of the economy (waste management, modernization of the industrial sector).	[106,107]
			X	1. National Bioeconomy Strategy–Costa Rica 2020–2030 (2020).	1. Sustainable production with high added value-sustainable use of resources-circular use of biomass—biotechnological progress.	[108]
Colombia	X			1. National Policy for Integrated Solid Waste Management 2016–2030 (2016). 2. Law 1990/2019. 3. Resolution No. 1407/2018.	1. Integrated solid waste management-promotion of CE, sustainable development, climate change adaptation and mitigation. 2. Prevention of food loss and waste. 3. Post-consumption of packaging. Responsibility of producers in the management of this waste.	[109,110,111]
		X		1. National Circular Economy Strategy 2018–2022.	1. New economic development model. Maximizing the value of resources-closing material cycles—Development of new business models-Industrial symbiosis.	[112]
			X	1. Bioeconomy for a living and diverse Colombia: Towards a knowledge-driven society (2020). 2. Colombia Green Growth Roadmap (2018). 3. National Program for Sustainable Biotrade (PNBS) 2014–2024 (2014). 4. Policy for the Commercial Development of Biotechnology from the Sustainable Use of Biodiversity (2011).	1. National strategy dedicated to the development of the bioeconomy in Colombia. 2. New sources of sustainable growth-Supply of natural capital to produce environmental goods and services. 3. Development of value chains based on the shared management of natural resources. 4. Development of economic, technical, institutional, and legal conditions to attract public and private resources and create enterprises and products based on the sustainable use of biodiversity.	[113,114,115,116]
Cuba	X			1. Ministry of Industries Policy on Increased Recycling of Raw Materials (2012). Updated in 2014.	1. Promotes recycling through a new management approach based on economic incentives, instruments, and new management models.	[117]
México	X			1. National Zero Waste Vision (2019). 2. General Law for the Prevention and Comprehensive Management of Wastes (2003).	1. Transform the traditional waste management model into a CE model. 2. Guarantee the right to a healthy environment and promote sustainable development by adequately managing hazardous waste and MSW.	[118,119]
		X		1. Circular Economy Action Plan.	1. Zero waste through a series of strategies such as reducing the amount of packaging, regulation for the reduction of single-use products, and proper waste management, among others.	[120]
			X	2. Agreement approving and publishing the update of the Transition Strategy to Promote the Use of Cleaner Technologies and Fuels (2020).	2. To regulate the sustainable use of energy, obligations in terms of clean energy, and the reduction of polluting emissions.	[121]

WM: Waste Management, CE: Circular Economy, B/CB: Bioeconomy/Circular Bioeconomy, OR: Other related. R: References. Table prepared by the authors based on the strategies published in [23]. The instrument can be framed in the selected approach, which is indicated by X.

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
