# Peer review of "Management of the Organic Fraction of Municipal Solid Waste in the Context of a Sustainable and Circular Model: Analysis of Trends in Latin America and the Caribbean"

_ijerph, 2022, doi:10.3390/ijerph19106041_

Round 1
Reviewer 1 Report
The paper has improved substantially. However, three issues remain:
1) I do not agree that there are "little amount of papers addressing the implications in human health of inadequate management of MSW." Your decision to delete the phrase “that affect the population’s health” from the manuscript, will definitely lower the quality of the manuscript. Both the topic and the journal are pretty much related to public health and the environment. I still suggest adding a paragraph (or some sentences) on the issues I raised by reviewing the literature, including the World Health Organization reports.
2) Also, please enlarge Figure 3 and Figure 4 to span from the left to right margins, similar to Tables 2 and 3. This will make the bibliometric analysis results to be more legible. This is important.
3) There is no need for another section for implications. They should be part of the conclusion section.
Author Response
Journal: International Journal of Environmental Research and Public Health
Manuscript number: 1698089
Article type: Review
Dear reviewer,
We appreciate the time and effort that you have dedicated to providing your valuable feedback on our manuscript. We carefully revised your suggestions and here we respond point-by-point each one. In the manuscript, we already made the respective changes.
Reviewer comments:
Reviewer 1
Comments and Suggestions for Authors
The paper has improved substantially. However, three issues remain:
1) I do not agree that there are “little amount of papers addressing the implications in human health of inadequate management of MSW.” Your decision to delete the phrase “that affect the population’s health” from the manuscript, will definitely lower the quality of the manuscript. Both the topic and the journal are pretty much related to public health and the environment. I still suggest adding a paragraph (or some sentences) on the issues I raised by reviewing the literature, including the World Health Organization reports. Lines 64 – 78.
R// We appreciate your insightful suggestions. We made the respective changes in the manuscript. Unfortunately, we had misunderstood the previous comment; thank you for clarifying this matter. We have reinforced the idea you expressed, adding a paragraph to the Introduction section with information we found in WHO reports 2007 and 2015. Also, some relevant studies and reviews that have been carried out to report the effects of the inadequate MSW management on human health.
2) Also, please enlarge Figure 3 and Figure 4 to span from the left to right margins, similar to Tables 2 and 3. This will make the bibliometric analysis results to be more legible. This is important.
R// We made the respective changes in the figures, improving the image quality and size.
3) There is no need for another section for implications. They should be part of the conclusion section.
R/ Thank you for your suggestion. We had modified the conclusions, including some theoretical and practical implications within the text.

This manuscript is a resubmission of an earlier submission. The following is a list of the peer review reports and author responses from that submission.
Round 1
Reviewer 1 Report
- Line 21: Please define LILACS at first mention
- Line 22: Please indicate the main research approaches used?
- Lines 36-37: what is the source of the first sentence of the Introduction section?
- Lines 36-54: there is little information on the implications of inadequate management of MSW on human health and socioeconomic development. For example, air pollution (foul odor and smoke from incineration) can cause respiratory tract and lung infections, water pollution (surface water contamination & groundwater contamination by leachate) that can result in cholera & typhoid. Also, it can obstruct stormwater flow resulting in floods, and disease vectors (insects and rodents) can spread malaria, dengue fever, and other communicable diseases. It is associated with exacerbating inequality because people living around garbage dumps and landfills experience poor health, low school attendance, inadequate services, and high rates of poverty and crimes more than in other parts of the city, and so on.
- Line 65: what are those “more significant” efforts?
- Lines 104-114: the novelty of the present study is not strongly justified. There is the need to highlight the research gap (holes in the literature) by identifying the limitations of similar studies (e.g., 2, 14, 15, and 16) in addressing the objectives of the present study.
- Lines 118-121: there is the need to justify selecting the LILACS and Scopus databases and limiting the documents to (journal?) articles only.
- The choice of the research method (bibliometric analysis) should also be justified and cite examples of studies that employed the method.
- Lines 165-179: what about the evolution of the topics? What was the initial and recent focus of the research?
- Section 3.1.3: it seems that, except for three studies, there is a correlation between the number of citations and the date of publication.
- Line 219: Q1 is 75th percentile and above (top 25%), but Q2 is between 50th to 74th percentile
- Figure 3 is not visible. Please separate B from A and enlarge both images to span from left to right, similar to Tables 2 & 3.
- Line 256-257: which is in the third and fourth place, 17.4% or 20%? Please check the order.
- Lines 458-459: I cannot identify IOR and POD acronyms and the “Time from Received to Published” from Table 4. Also, ** in the first row of Table 4 is not defined.
- Section 4: the conclusion section should underscore the implications of the study for urban waste management policies and practices.
Comments and Suggestions for Authors
- Line 21: Please define LILACS at first mention.
R/ We made the respective changes in the manuscript
- Line 22: Please indicate the main research approaches used?
R/ We made the respective changes in the manuscript
- Lines 36-37: what is the source of the first sentence of the Introduction section?
R/ We made the respective changes in the manuscript
- Lines 36-54: there is little information on the implications of inadequate management of MSW on human health and socioeconomic development. For example, air pollution (foul odor and smoke from incineration) can cause respiratory tract and lung infections, water pollution (surface water contamination & groundwater contamination by leachate) that can result in cholera & typhoid. Also, it can obstruct stormwater flow resulting in floods, and disease vectors (insects and rodents) can spread malaria, dengue fever, and other communicable diseases. It is associated with exacerbating inequality because people living around garbage dumps and landfills experience poor health, low school attendance, inadequate services, and high rates of poverty and crimes more than in other parts of the city, and so on.
R// You are right, through the literature search we found little amount of papers addressing the implications in human health of inadequate management of MSW. This is we decided to delete the phrase “that affect the population’s health” from the manuscript.
- Line 65: what are those “more significant” efforts?
R/ We made the respective changes in the manuscript, adding the requested information.
- Lines 104-114: the novelty of the present study is not strongly justified. There is the need to highlight the research gap (holes in the literature) by identifying the limitations of similar studies (e.g., 2, 14, 15, and 16) in addressing the objectives of the present study.
R/ Thank you for the suggestion. We made the respective changes in the manuscript
“This study aims to address the identified research gap in previous studies, which includes a lack of evaluation of scientific production on OFMSW, identifying the main approaches of the studies and offering a scheme of the regulatory framework in LAC to support SCPMs, along with the current barriers.”
- Lines 118-121: there is the need to justify selecting the LILACS and Scopus databases and limiting the documents to (journal?) articles only.
R/ Thank you for calling our attention into this. We had included the justifications and made the respective changes in the manuscript
- The choice of the research method (bibliometric analysis) should also be justified and cite examples of studies that employed the method.
R/ We made the respective changes in the manuscript
- Lines 165-179: what about the evolution of the topics? What was the initial and recent focus of the research?
R/ We made the respective changes in the manuscript
- Section 3.1.3: it seems that, except for three studies, there is a correlation between the number of citations and the date of publication.
- Line 219: Q1 is 75th percentile and above (top 25%), but Q2 is between 50th to 74th percentile.
R/ Thank you for noticing this, we had a mistake. We made the respective changes in the manuscript
- Figure 3 is not visible. Please separate B from A and enlarge both images to span from left to right, similar to Tables 2 & 3.
R/ We made the respective changes in the figures, we think now are more visible. Regarding the tables we followed the journals instructions
13 Line 256-257: which is in the third and fourth place, 17.4% or 20%? Please check the order.
R/ We made the respective changes in the manuscript, we had misplaced the information.
- Lines 458-459: I cannot identify IOR and POD acronyms and the “Time from Received to Published” from Table 4. Also, ** in the first row of Table 4 is not defined.
R/ We made the respective changes in the table
- Section 4: the conclusion section should underscore the implications of the study for urban waste management policies and practices.
R/ This is an important remark. We had included section 5: Theoretical and practical implications.
Reviewer 2 Report
Dear Authors
this is interesting material, thank you for that. I think it is valuable input and may be published with such prerogatives:
even as the work targets the LA and Carribean, I would suggest in Introduction part mention such authors that have been doing intensive work on topics related to linked themes missing in the region. One of such is landfill mining and the requirement is to look through these paradigmatic works and cite them in references:
e.g. Hogland et al and Krook et al
Otherwise the topic is covered quiet well
Comments and Suggestions for Authors
Dear Authors
this is interesting material, thank you for that. I think it is valuable input and may be published with such prerogatives:
even as the work targets the LA and Carribean, I would suggest in Introduction part mention such authors that have been doing intensive work on topics related to linked themes missing in the region. One of such is landfill mining and the requirement is to look through these paradigmatic works and cite them in references:
e.g. Hogland et al and Krook et al
Otherwise the topic is covered quiet well
R/ Thank you for acknowledging the importance of this work. You are right and landfill mining is a missing topic in the region, but it is important indeed to provide some information to the readers. We had included two references from the suggested author aiming the topic:
https://doi.org/10.1016/j.wasman.2011.10.015
https://doi.org/10.1007/s10163-017-0683-4
Reviewer 3 Report
The paper was revised according to the journal rules.
Few revisions are required and they are reported below:
- please add a nomenclature list with all acronyms and parameters used
- the introduction section should be improved, adding:
- specifications on trace compounds from OFMSW, citing sutdies such as (10.1016/j.wasman.2021.09.041)
- improve this section adding also the environmental effects, especially on GWP aspects
- section 2 should be more clear
- revise figure 3
- revise figure 4
- a graphical abstract should be added
Comments and Suggestions for Authors
The paper was revised according to the journal rules.
Few revisions are required and they are reported below:
- please add a nomenclature list with all acronyms and parameters used
R/ Thank you for this recommendation. We did not include the list of acronyms according to the journals instructions, which states: “Acronyms/Abbreviations/Initialisms should be defined the first time they appear in each of three sections: the abstract; the main text; the first figure or table. When defined for the first time, the acronym/abbreviation/initialism should be added in parentheses after the written-out form.”
- the introduction section should be improved, adding:
- specifications on trace compounds from OFMSW, citing sutdies such as (10.1016/j.wasman.2021.09.041)
R/ Thank you for the suggestion, this is really good research. We had included the reference you advised.
- improve this section adding also the environmental effects, especially on GWP aspects
R/ Thank you for the recommendation. We made the respective changes in the manuscript
- section 2 should be more clear
R/ We made the respective changes in the manuscript following the recommendations from all reviewers in this matter.
- revise figure 3
R/ We made the respective changes in the figure
- revise figure 4
R/ We made the respective changes in the figure
- a graphical abstract should be added
R/ We included the graphical abstract
Reviewer 4 Report
The manuscript ID ijerph-1604216 entitled “Management of the organic fraction of municipal solid waste in the context of a sustainable and circular model. Analysis of trends in Latin America and the Caribbean” is an interesting and valuable study. The Authors carefully analysed the available literature related to the analyse the most relevant aspects of the management of the organic fraction of municipal solid waste and the sustainable and circular production models in Latin America and the Caribbean. The analysis of 190 studies that were obtained from the Scopus and LILACS databases and finally use of 104 literature items is very impressive.
The title is clear. The content is in accord with title. The size of the article is appropriate to the contents. In the introduction it is clearly described the state of the art of the investigated problem. Methodology used is appropriate. The paper was written in standard, grammatically correct English, small corrections are necessary.
The Authors properly present the current scientific achievements in the field of synthesis of these important form ecologically and environmentally topic. The authors effectively try to analyse and summarize the current state of advancement of different types of technology and the possibilities of their application in practice, and on this basis present the potential and prospects of their future use. The manuscript is written in the correct language, its layout and the extracted chapters are logical. The presented content corresponds to the International Journal of Environmental Research and Public Health profile.
I believe that the manuscript may be a valuable work, it is a valuable compendium of scientific knowledge. I believe that it may be published in current form.
Good luck!
Comments and Suggestions for Authors
The manuscript ID ijerph-1604216 entitled “Management of the organic fraction of municipal solid waste in the context of a sustainable and circular model. Analysis of trends in Latin America and the Caribbean” is an interesting and valuable study. The Authors carefully analysed the available literature related to the analyse the most relevant aspects of the management of the organic fraction of municipal solid waste and the sustainable and circular production models in Latin America and the Caribbean. The analysis of 190 studies that were obtained from the Scopus and LILACS databases and finally use of 104 literature items is very impressive.
The title is clear. The content is in accord with title. The size of the article is appropriate to the contents. In the introduction it is clearly described the state of the art of the investigated problem. Methodology used is appropriate. The paper was written in standard, grammatically correct English, small corrections are necessary.
The Authors properly present the current scientific achievements in the field of synthesis of these important form ecologically and environmentally topic. The authors effectively try to analyse and summarize the current state of advancement of different types of technology and the possibilities of their application in practice, and on this basis present the potential and prospects of their future use. The manuscript is written in the correct language, its layout and the extracted chapters are logical. The presented content corresponds to the International Journal of Environmental Research and Public Health profile.
I believe that the manuscript may be a valuable work, it is a valuable compendium of scientific knowledge. I believe that it may be published in current form.
Good luck!
R/ Thank you for acknowledging the importance of this work. One of our purposes is to make novel and significant contributions to the scientific literature providing to the researchers and decision makers in Latin America and the Caribbean helpful evidence on the relevance that policies on SCPMs have gained in recent years and their influence on the development of research related to the characterisation and alternatives for the reduction and/or valorization of waste.
Thank you in advance for your time and attention,
Yours faithfully,
Leidy Marcela Ulloa-Murillo.